# UNDERSTANDING DEEPRESEARCH VIA REPORTS

## ABSTRACT

DeepResearch agents represent a transformative AI paradigm, conducting expert-level research through sophisticated reasoning and multi-tool integration. However, evaluating these systems remains critically challenging due to open-ended research scenarios and existing benchmarks that focus on isolated capabilities rather than holistic performance. Unlike traditional LLM tasks, DeepResearch systems must synthesize diverse sources, generate insights, and present coherent findings, which are capabilities that resist simple verification. To address this gap, we introduce DEEPRESEARCH-REPORTEVAL, a comprehensive framework designed to assess DeepResearch systems through their most representative outputs: research reports. Our approach systematically measures three dimensions: quality, redundancy, and factuality, using an innovative LLM-as-a-Judge methodology achieving strong expert concordance. We contribute a standardized benchmark of 100 curated queries spanning 12 real-world categories, enabling systematic capability comparison. Our evaluation of four leading commercial systems reveals distinct design philosophies and performance trade-offs, establishing foundational insights as DeepResearch evolves from information assistants toward intelligent research partners. We will release the prompts and data we use.

## 1 INTRODUCTION

DeepResearch agents represent a new class of AI systems designed to tackle knowledge-intensive research tasks. Traditional Large Language Models (LLMs) excel at conversational interactions but struggle with research applications. These applications demand systematic evidence collection, rigorous validation, comprehensive synthesis, and transparent reasoning trails. DeepResearch addresses these challenges by simulating human research processes through agent interactions. The system clarifies research intent and collects evidence from sources. It validates information across modalities and synthesizes findings into comprehensive reports. These reports serve as the primary output format that delivers research results to users. This enables completion of complex research tasks in minutes rather than hours, spanning domains such as finance, science, and engineering.

However, the sophistication of DeepResearch agents raises critical evaluation challenges. Since reports constitute the primary form of generated content, they serve as the main interface for assessing system capabilities. How do we assess automatically generated research report quality? What metrics capture task requirements through these outputs? Traditional evaluation approaches (Bai et al., 2024; Wu et al., 2025b; Paech, 2023) designed for simpler text generation prove inadequate. They cannot effectively assess research-oriented agent capabilities through outputs. Existing benchmarks (Mialon et al., 2023; Trivedi et al., 2022; Wei et al., 2025) focus on search strategies or multi-hop reasoning but fail to capture the holistic quality demands of comprehensive research outputs. These limitations stem from theirs scope: they evaluate isolated capabilities rather than integrated performance of end-to-end research systems. Current evaluation methods lack sophistication to assess how systems transform user queries into actionable research objectives, synthesize information from diverse sources, or produce coherent insights with evidence grounding.

To better understand DeepResearch systems through report analysis, we first examine the key stages that define their operational framework. Current leading DeepResearch systems are rapidly evolving (Google, 2025; OpenAI, 2025; Qwen, 2025; Perplexity, 2025) to bridge user information needs with real-world research complexity. Understanding the fundamental three-stage architecture is crucial for evaluating capabilities, as shown in Figure 1:

Figure 1: DeepResearch systems operate through: Interaction, Investigation, and Synthesis.

- **Interaction**. This stage transforms vague user queries into actionable research objectives through intelligent clarification. Users submit initial queries with potential multi-modal files. The system refines understanding through active questioning, ensuring alignment with user intent.
- **Investigation**. The operational core deploys advanced reasoning, planning, and tool usage for comprehensive investigation. The system maintains coherent direction while efficiently exploring diverse sources, managing memory seamlessly as the investigation unfolds.
- **Synthesis**. This stage synthesizes findings into coherent, well-grounded insights. Outputs include structured reports with charts, tables, and images, plus references for traceability. Alternative formats may encompass webpages and podcasts.

In this paper, we propose to understand DeepResearch systems via reports, as reports represent the most classical and representative form of DeepResearch outputs. High-quality research reports exhibit clear structure, rigorous logic, dense information content, and authentic citations, all of which are characteristics essential for knowledge-intensive research scenarios. We introduce the DEEPRESEARCH-REPORTEVAL framework, a hybrid evaluation methodology that employs LLM-as-a-Judge to assess the quality of the final report, combined with expert human judgment to ensure reliability. The framework evaluates report quality across multiple dimensions including comprehensiveness, redundancy, and factuality. We will release a curated dataset comprising 100 queries spanning diverse categories with 100 corresponding reports, facilitating the systematic evaluation.

## 2 THE FRAMEWORK OF DEEPRESEARCH-REPORTEVAL

In this section, we present the DEEPRESEARCH-REPORTEVAL framework for evaluating Deep-Research systems. We first explain why research reports provide an effective evaluation medium for understanding DeepResearch capabilities (Section 2.1), identify the core challenges inherent in report-based evaluation (Section 2.2), and detail our proposed evaluation methodology (Section 2.3).

### 2.1 WHY EVALUATE DEEPRESEARCH AGENTS THROUGH REPORT QUALITY?

**Beyond Traditional Search Tasks.** DeepResearch systems fundamentally differ from traditional AI tasks in their scope and complexity. While existing benchmarks like BrowseComp (Wei et al., 2025) and HotpotQA (Yang et al., 2018) evaluate systems' ability to retrieve specific facts through multi-hop reasoning, DeepResearch tasks require comprehensive analysis and synthesis across multiple information sources. As illustrated in Figure 2, traditional search queries seek precise, verifiable answers that can be evaluated through exact match, whereas DeepResearch queries demand thorough investigation and nuanced insights that resist simple factual verification.

**Real-World Usage Patterns.** To understand the breadth of DeepResearch applications, we analyzed over 150K real-world queries. Our LLM-based classification reveals twelve distinct categories,

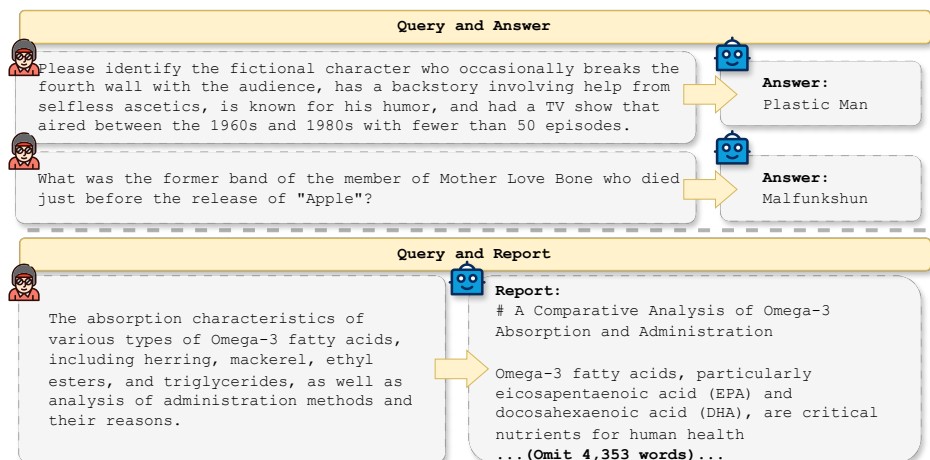

Figure 2: Current search queries (upper), from BrowseComp (Wei et al., 2025) and HotpotQA (Yang et al., 2018), seek specific answers through multi-hop reasoning. DeepResearch queries (lower) demand comprehensive investigation and synthesis, producing detailed analytical reports as shown in this figure. Additional examples appear in Appendix A.

with *Science & Technology* (37.3%) and *Economy & Business* (17.2%) dominating usage patterns (Figure 3). These domains naturally gravitate toward systematic research methodologies, requiring structured analysis of complex, evolving information landscapes where decisions depend on comprehensive understanding rather than isolated facts.

**Reports as Natural Evaluation Medium.** The system's deployment across diverse domains—from *Health & Medicine* to *History & Culture*—demonstrates both its versatility and the evaluation challenges this creates. Unlike narrow-domain tasks with clear success metrics, DeepResearch must synthesize information, generate novel insights, and present findings coherently across vastly different knowledge areas. Given this complexity, research reports emerge as the natural evaluation medium: they capture the system's end-to-end capabilities in information gathering, analysis, synthesis, and presentation. This motivation drives our DEEPRESEARCH-REPORTEVAL framework design.

## 2.2 WHAT MAKES DEEPRESEARCH REPORT EVALUATION CHALLENGING?

**Beyond Traditional Writing Assessment.** Evaluating DeepResearch reports presents unique challenges beyond ordinary long-form writing evaluation (Que et al., 2024; Bai et al., 2024; Wu et al., 2025b; Paech, 2023). Traditional writing assessments focus on structural clarity, length control, or stylistic requirements. DeepResearch emphasizes knowledge-oriented outputs with rigorous research methodology, factual evidence integration, and analytical insights.

**Assessing Comprehensiveness and Analytical Depth.** DeepResearch report length varies organically with query complexity and available information. It does not follow predefined constraints.

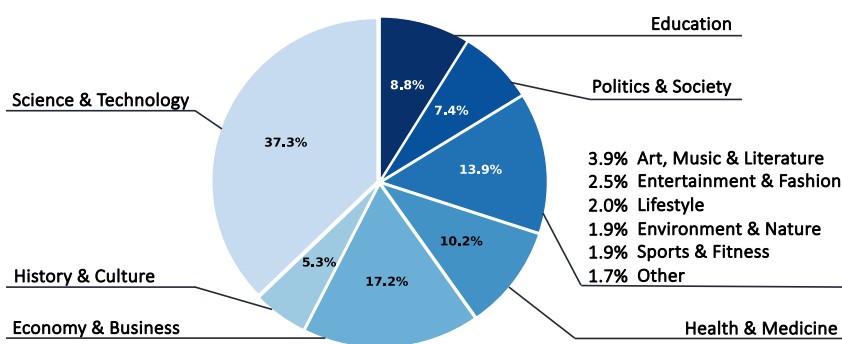

Figure 3: Visualization of category distribution of DeepResearch queries.

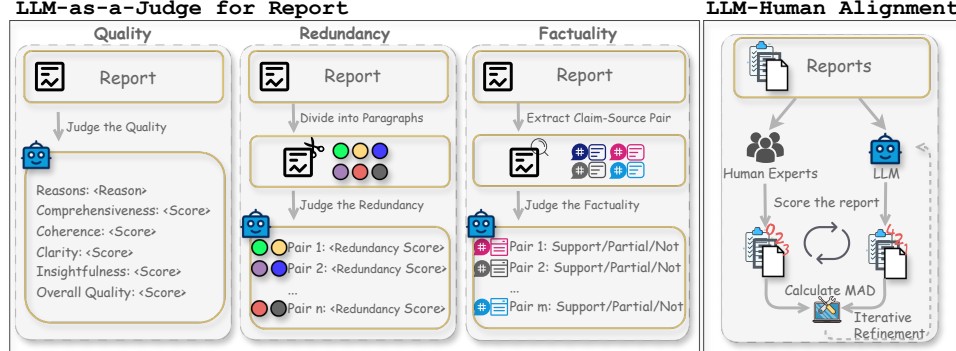

Figure 4: Overview of the DEEPRESEARCH-REPORTEVAL framework. The LLM-as-a-Judge approach is used to evaluate reports along the dimensions of quality, redundancy, and factuality, while LLM–Human alignment is employed to ensure the reliability.

Content richness must match research scope, requiring comprehensive coverage across multiple dimensions. Unlike tasks such as LongBench-Write (Bai et al., 2024) that emphasize length control, evaluation must assess thoroughness and analytical depth rather than adherence to word counts.

**Ensuring Source Reliability and Factuality.** Report generation requires grounding in factual evidence with accurate source summarization and citation. This distinguishes it from creative writing tasks like WritingBench (Wu et al., 2025b). Critical evaluation dimensions include citation accuracy, source reliability, fact consistency, and avoiding contradictory evidence. Factuality assessment must examine both the credibility of sources and the accuracy of their representation. However, reports that merely enumerate facts without logical reasoning provide limited value. High-quality outputs must transcend factual correctness to deliver meaningful interpretations and insights.

Based on these considerations, we propose the DEEPRESEARCH-REPORTEVAL framework to comprehensively evaluate DeepResearch agentic capabilities through systematic report assessment. Our framework addresses the fundamental challenge of measuring analytical quality and evidence utilization effectiveness in knowledge-intensive research tasks. Unlike DeepResearchBench (Du et al., 2025), which relies on comparative evaluation against predefined reference reports, our approach enables direct assessment of individual reports. We evaluate based on their comprehensive coverage and methodological rigor.

## 2.3 HOW DO WE EVALUATE DEEPRESEARCH REPORTS?

The DEEPRESEARCH-REPORTEVAL framework centers on comprehensive assessment of report quality and evidence utilization effectiveness. We decompose DeepResearch report evaluation into three essential dimensions: **Quality**, **Redundancy**, and **Factuality**. The goal is to produce evaluation scores that align closely with human expert judgments. Throughout the evaluation design, human experts are involved to ensure that the results generated by our framework are reliable and credible. For this work, we focus on textual content of reports, including text and tables, reserving visual elements such as images and charts for future investigation. Figure 4 provides an overview of DEEPRESEARCH-REPORTEVAL, which we detail throughout this section.

### 2.3.1 QUALITY

DeepResearch systems generate comprehensive research reports across diverse domains. Effective evaluation requires systematic assessment of their linguistic and structural qualities. We establish a five-dimensional framework capturing essential aspects of report excellence. This framework addresses both fundamental requirements and advanced analytical capabilities that determine effectiveness in communicating knowledge and insights.

- **Comprehensiveness** evaluates whether reports provide complete topic coverage without major omissions. Quality reports address key aspects systematically. They maintain balanced discussions that thoroughly explore relevant dimensions, avoiding superficial treatment of important subtopics.

- **Coherence** assesses organizational structure and logical flow throughout the document. Well-structured reports present ideas in clear sequences. Sections and subsections connect smoothly. This guides readers naturally from introduction through analysis to conclusions.

- **Clarity** examines language fluency, accuracy, and stylistic consistency across all paragraphs. Professional reports employ precise vocabulary and maintain consistent tone. They eliminate redundancy and avoid awkward expressions. This prevents impediments to reader comprehension.

- **Insightfulness** determines whether reports transcend mere information compilation. Exceptional reports demonstrate deep subject understanding. They present fresh analytical viewpoints and provide well-reasoned arguments. This reflects independent critical thinking beyond basic summarization.

- **Overall Quality** synthesizes impressions from all evaluation criteria into a holistic assessment. This comprehensive judgment captures the report's general effectiveness. It considers how individual dimensions combine successfully. The result creates cohesive, impactful documents.

For each report, we employ the LLM-as-a-Judge approach to systematically evaluate quality across multiple dimensions, assigning scores from 0 to 4 based on carefully designed prompt templates detailed in Appendix B.1. To ensure rigorous assessment, we require the model to provide detailed justifications for each score, promoting deliberate and transparent evaluation. This framework enables us to assess whether reports meet fundamental standards for readability, clarity, and expressiveness.

### 2.3.2 REDUNDANCY

In DeepResearch systems, reports are generated by synthesizing information from diverse sources into a coherent narrative. However, these systems frequently exhibit over-dependence on limited sources, recycling their content across sections with superficial modifications in phrasing or analytical framing. Consequently, multiple paragraphs or subsections develop thematic overlap or outright duplication, artificially inflating report length while contributing minimal substantive insight. This redundancy undermines reading efficiency and obscures key arguments, ultimately compromising analytical depth and degrading overall report quality. Standard LLM-as-a-Judge evaluation approaches often fail to detect these subtle redundancy patterns when assessing complete reports. To overcome this limitation, we develop a specialized methodology for identifying and measuring content redundancy.

Since redundancy often manifests as repeated content or overlapping discussions across different sections of a DeepResearch report, we formalize the detection task as a systematic LLM-as-a-Judge workflow. The comprehensive procedure consists of three main steps.

**Paragraph Segmentation**. We divide the report into paragraph-level units. Let the input report be denoted as $r = (p_1, p_2, \ldots, p_k)$, where $p_i$ is the $i$-th paragraph and $k$ is the total number.

**Pairwise Redundancy Assessment**. We create all possible $n$ paragraph pairs Pair $= (p_i, p_j)$ and use the LLM-as-a-Judge approach to assess redundancy for each pair. The model determines whether the paragraphs are semantically independent, with no repeated viewpoints, examples, or expressions. A score from 0 to 4 is assigned, where 4 indicates no redundancy. The redundancy score is denoted by $\text{Score}_R(\text{Pair})$. The prompt template is in Appendix B.2.

**Overall Score Calculation**. The overall redundancy score $\text{Score}_R(r)$ represents the report's redundancy level. It is defined as the arithmetic average of redundancy scores of all possible pairs:

$$\text{Score}_R(r) = \frac{1}{n} \sum_{i=1}^{n} \text{Score}_R(\text{Pair}_i). \tag{1}$$

This approach precisely detects semantic overlap between paragraphs in DeepResearch reports and provides a comprehensive framework for measuring content redundancy in generated documents.

### 2.3.3 FACTUALITY

As discussed in Section 2.2, DeepResearch Reports are grounded in factual evidence and include extensive citations to substantiate their arguments. To verify whether claims are genuinely supported by cited sources and identify potential hallucinations, we conduct comprehensive factuality evaluation.

**Claim-Source Alignment Assessment**. We assess the alignment between each claim and its associated cited source through a systematic process. First, we extract all claims $c_i$ from the generated

Table 1: MAD values obtained using DEEPRESEARCH-REPORTEVAL and human experts. For quality, the MAD value represents the average across five dimensions: comprehensiveness, coherence, clarity, insightfulness, and overall quality. The factuality MAD value is computed using the average support score of each report. The range of MAD values for quality and redundancy is $0$ to $4$, while the range for factuality is $0$ to $2$.

| Quality MAD | Redundancy MAD | Factuality MAD |
|:-----------:|:--------------:|:--------------:|
| 0.72        | 0.31           | 0.29           |

report along with their corresponding cited sources $s_i$. Then, for each claim-source pair $(c_i, s_i)$, we employ the LLM-as-a-Judge approach to determine whether source $s_i$ semantically and logically supports claim $c_i$. When a claim cites multiple sources, we evaluate it independently against each source. The support score $\text{Score}_F(c_i, s_i)$ follows three-level classification: $1$ indicates full source support, $0$ indicates partial support, and $-1$ indicates no support. The prompt is in Appendix B.3.

**Automated Factuality Measurement**. We introduce two complementary metrics to quantify the factuality score of an entire report. The first metric, *Average Support Score*, calculates the mean of all support scores, capturing overall alignment strength between claims and cited sources:

$$\text{Score}_{F1}(r) = \frac{1}{m} \sum_{i=1}^{m} \text{Score}_F(c_i, s_i), \tag{2}$$

where $m$ denotes the total number of claim-source pairs in the report. The second metric, *Strong Support Rate*, measures the proportion of claim-source pairs that achieve full support, providing insight into the reliability of well-substantiated claims:

$$\text{Score}_{F2}(r) = \frac{1}{m} \left| \text{Score}_F(c_i, s_i) = 1 \right|, \tag{3}$$

where $|\cdot|$ denotes the count of pairs scoring $1$. These metrics together enable comprehensive and automated assessment of factual accuracy, significantly reducing reliance on manual annotation while providing nuanced evaluation of report factuality.

### 2.3.4 LLM-HUMAN ALIGNMENT PROCESS

Evaluating DeepResearch reports presents significant challenges due to their extensive length and the need for multi-dimensional assessment across criteria that lack clear evaluation standards. In this work, we evaluate reports across three critical aspects: quality, redundancy, and factuality, with the goal of ensuring that LLM-as-a-Judge assessments closely align with human expert judgments.

To achieve this alignment, we design an iterative prompt refinement mechanism (Yuksekgonul et al., 2024) to perform LLM-Human alignment, illustrated on the right side of Figure 4. We first collect 120 queries and generate corresponding reports, then recruit three human experts to score all reports according to previously defined guidelines for quality, redundancy, and factuality. Detailed criteria ensure consistent scoring. Subsequently, we iteratively refine the prompt templates used for LLM-as-a-Judge evaluation through manual adjustments, ensuring that the model's scores for the 120 reports closely match average scores provided by experts.

We quantify the alignment between LLM scores and human expert judgments using Mean Absolute Deviation (MAD), which measures the average absolute difference between individual data points and their mean value. This metric provides robust assessment of scoring consistency across evaluators. Formally, let $R = \{r_1, r_2, \ldots, r_N\}$ represent the set of reports. Using redundancy as example, let $\text{Score}_R(r_i)$ denote the LLM-generated redundancy score for report $r_i$, and $\text{Score}_R^H(r_i)$ represent the averaged score from three human experts for the same report. The redundancy MAD is calculated as:

$$\text{MAD}_R = \frac{1}{N} \sum_{i=1}^{N} \left| \text{Score}_R(r_i) - \text{Score}_R^H(r_i) \right|, \tag{4}$$

where $N = 120$ is the number of evaluated reports. Lower MAD values indicate stronger agreement between LLM-generated scores and human expert judgments. This reflects more reliable automated evaluation. The iterative refinement process continues until the LLM's scoring demonstrates robust

Table 2: Statistics for 100 generated reports from each DeepResearch system. We report averages and standard deviations for report length (tokens), redundancy (paragraph pairs, capped at 30 due to quadratic scaling), and factuality (claim–source pairs).

| | OpenAI | Perplexity | Gemini | Qwen |
|---|---|---|---|---|
| Report Length | $6917.6_{\pm 4319.7}$ | $1245.2_{\pm 720.0}$ | $9234.5_{\pm 4345.1}$ | $5466.8_{\pm 778.3}$ |
| Paragraph Pair Count | $25.5_{\pm 4.5}$ | $20.4_{\pm 8.8}$ | $29.5_{\pm 2.9}$ | $29.3_{\pm 3.2}$ |
| Claim-Source Pair Count | $109.3_{\pm 46.9}$ | $24.7_{\pm 32.1}$ | $78.3_{\pm 35.4}$ | $116.5_{\pm 34.7}$ |

Table 3: Overall evaluation results of different DeepResearch systems on 100 queries from DEEPRESEARCH-REPORTEVAL. Quality and redundancy use a 0–4 scale. For factuality, average support score ranges from $-1$ to 1, while strong support rate is percentage-based. **Bold** indicates best performance, underline indicates second-best, with higher values representing better results.

| | | OpenAI | Perplexity | Gemini | Qwen |
|---|---|---|---|---|---|
| | Comprehensiveness | 3.57 | 3.16 | 3.65 | **3.80** |
| | Coherence | 3.29 | **3.60** | 3.15 | 3.40 |
| Quality | Clarity | 3.43 | 3.46 | **3.50** | 3.33 |
| | Insightfulness | 3.01 | 2.96 | 3.22 | **3.38** |
| | Overall Quality | 3.28 | 3.07 | 2.93 | **3.54** |
| Redundancy | Overall Redundancy | 3.52 | **3.71** | 3.15 | 3.50 |
| Factuality | Average Support Score | 0.49 | 0.42 | 0.46 | **0.55** |
| | Strong Support Rate | **0.71** | 0.56 | 0.55 | 0.69 |

alignment with human evaluation preferences. Table 1 presents the MAD values obtained using our final optimized prompts. These values are consistently small across all dimensions, demonstrating that our LLM-as-a-Judge methodology achieves substantial agreement with human experts.

To further validate that DEEPRESEARCH-REPORTEVAL's scoring aligns with human preferences, we conduct an additional ranking-based evaluation. We generate three reports for each of the 120 queries using different LLMs and ask human experts to rank these reports. Subsequently, DEEPRESEARCH-REPORTEVAL scores the same reports and produces rankings based on quality and redundancy scores. We evaluate alignment using exact match accuracy between human and model rankings, achieving 61.11% agreement. This result demonstrates that our iterative refinement mechanism effectively enhances both the credibility and alignment of LLM-based evaluations with human expert judgments.

## 3 EXPERIMENTAL EVALUATION

As described in Section 2.1, typical DeepResearch queries differ significantly from multi-step search queries, being more complex and open-ended. To better evaluate DeepResearch systems and provide the research community with evaluation data, we carefully create and release 100 queries within the DEEPRESEARCH-REPORTEVAL framework, each tagged with its category. The category distribution closely reflects real-world user behavior when interacting with DeepResearch systems, as shown in Figure 3. These queries range from detailed, specific research requirements to broader, open-ended inquiries, mirroring how users engage with systems. Example queries are in Appendix A.

**Experimental Setup**. Most open-source DeepResearch frameworks are based on complex workflow designs (Zhang et al., 2025; Hu et al., 2025; ByteDance & contributors., 2025; LangChain & contributors., 2025), making fair comparison difficult. Therefore, we select four commercial DeepResearch systems: OpenAI (OpenAI, 2025), Perplexity (Perplexity, 2025), Gemini (Google, 2025), and Qwen (Qwen, 2025), which are typically more mature and offer stable performance. For the first three systems, we use their web interfaces, manually generating and downloading one report for each query. All LLM-as-a-Judge evaluations use GPT-4o, with both queries and reports in English. Report collection was conducted between August and early September. As these commercial systems are continuously updated, our experimental results should be interpreted with this limitation in mind.

**Evaluation Metrics and Statistics**. Table 2 presents statistics for 100 generated reports from each DeepResearch system. Report length is measured in tokens. For redundancy evaluation, we count paragraph pairs, with a maximum of 30 due to quadratic scaling. We exclude the first and

last paragraphs from pairing, as these sections are typically introductory or concluding, and some repetition is acceptable. For factuality evaluation, we count claim-source pairs. Table 3 presents the overall performance across multiple dimensions.

**Length vs. Quality Trade-offs**. As shown in the results, Perplexity's average report length is significantly shorter than the other three systems, at only 1245.2 tokens. This reflects a deliberate design philosophy prioritizing conciseness and readability. The system often adopts bullet-point structures that enhance user accessibility. Consequently, it achieves high scores in coherence and clarity, with low redundancy at 3.71. However, this approach creates tension with analytical depth. While it facilitates clear and visually organized information delivery, it sacrifices comprehensive analysis. This trade-off appears in its lowest scores for comprehensiveness and insightfulness.

**Finding the Sweet Spot in Report Length**. Reports generated by OpenAI and Qwen have average lengths of approximately 5000–7000 tokens. Gemini's reports are longer, averaging over 9000 tokens. This length variation reveals a fundamental challenge in research system design. Systems must balance comprehensive coverage with digestible presentation. Their quality scores are generally similar, with Qwen scoring slightly higher. This suggests that optimal report length may exist within a specific range. It does not follow a linear relationship with quality.

**Analytical Excellence: When Depth Meets Readability**. Qwen achieves the best performance in both comprehensiveness and insightfulness. This reflects the analytical depth of its reports. It demonstrates that sophisticated reasoning capabilities can be maintained while preserving readability. Gemini also performs well across individual quality dimensions but has the lowest overall quality score. This may partly be due to potential evaluation bias in the LLM-as-a-Judge process when assessing substantially longer reports. This phenomenon highlights a critical evaluation challenge. Longer content may face systematic penalties regardless of actual quality.

**Evidence-Based Credibility: The Foundation of Trust**. In terms of factuality, Qwen and OpenAI stand out as the most evidence-grounded systems. They establish a new standard for research system reliability. They score highly on both average support score and strong support rate. They also have the largest average number of claim–source pairs. This indicates a sophisticated ability to provide cited sources that substantiate the majority of claims in their reports. This suggests that effective fact-checking and source integration are becoming distinguishing factors among research systems.

# 4    REFLECTIONS ON DEEPRESEARCH

DeepResearch represents a rapidly evolving class of agent systems that has garnered substantial interest from the research community. While we have introduced our DEEPRESEARCH-REPORTEVAL framework and demonstrated its effectiveness through comprehensive report-based evaluation, several important questions and opportunities remain unexplored. This section presents open discussions on key challenges and future perspectives for advancing DeepResearch systems. Due to space limitations, we defer the discussion of additional related work to Appendix C.

## 4.1    THE ART OF QUERY REFINEMENT

**The Query-Quality Connection.** As presented in Section 1, before entering the research stage, the DeepResearch system typically engages in an interaction phase with the user. For example, in Qwen DeepResearch, the system asks several clarification questions, which serves as a critical foundation that cascades through the entire research pipeline. **Beyond Surface-Level Clarification.** User interaction enables the system to gather enriching details that improve subsequent research and report generation. However, what constitutes a good query remains underexplored and lacks systematic evaluation. A promising direction is enabling the LLM to proactively identify knowledge gaps, conflicting viewpoints, or unaddressed aspects, generating clarification questions with high informational value. Such strategic questioning could reduce uncertainty and guide research toward deeper insights, transforming the system from passive responder to active research collaborator.

## 4.2    RETHINKING SEARCH FOR RESEARCH

**The Paradigm Shift Challenge.** In the investigation stage of DeepResearch, the LLM must gather diverse supporting evidence for complex questions rather than seeking single direct answers, as

discussed in Section 2.1. Consequently, search agents that excel on traditional answer-centric benchmarks may prove ineffective in DeepResearch settings. This fundamental mismatch reveals a critical and problematic gap between existing search technologies and the nuanced requirements of research-oriented tasks.

**Toward Research-Centric Evaluation.** This necessitates rethinking search evaluation metrics and training objectives. New benchmarks for research-oriented retrieval are urgently needed, incorporating metrics such as evidence coverage, perspective diversity, cross-source consistency, and opposing viewpoints. Training should encourage agents to recall diverse relevant fragments rather than focusing solely on the most likely resource. This reflects the fundamental transition from *DeepSearch* to *DeepResearch*, where comprehensiveness and analytical depth take precedence over simple accuracy.

### 4.3 EXPANDING THE EVALUATION LANDSCAPE

**The Complexity of Holistic Assessment.** DeepResearch systems often involve very long reasoning chains and complex tool use, yet there is currently no widely accepted method for their evaluation. In this paper, we propose the DEEPRESEARCH-REPORTEVAL framework to assess generated reports as a way to evaluate the overall performance of a DeepResearch system, since reports are the most common and representative form of output and can directly reflect system quality. However, this end-to-end evaluation approach, while valuable, may obscure important intermediate processes and system behaviors that merit independent assessment.

**Multi-Dimensional Performance Metrics.** Beyond report evaluation, other metrics can capture different aspects of DeepResearch performance. For example, *search depth* measures the total number of webpages retrieved during the investigation process, indicating the comprehensiveness of information collection. Another useful metric is *end-to-end speed*, defined as the total time required for both research and final output generation. There is typically a trade-off between search depth and speed, as deeper searches often require more time. However, efficiency can be improved through parallel tool usage or collaboration among sub-agents. We advocate for developing a more diverse, practical, and comprehensive evaluation framework for DeepResearch systems within the community.

### 4.4 ENVISIONING THE DEEPRESEARCH AGENT

**Expanding Capabilities and Applications.** As a representative agent system, DeepResearch has attracted significant attention for its potential to support a wide range of impactful applications. As the system continues to evolve, we see several promising directions for future development. One direction is the integration of more customized tools, such as interfaces to private databases, which could unlock access to specialized knowledge repositories and proprietary datasets currently beyond the reach of web-based research.

**Toward Intelligent Research Partnerships.** Another transformative direction is enhancing interpretability by attaching confidence scores or dispute indicators to collected content, enabling users to make informed decisions about reliability of findings. A third direction is proactive feeding, where the system conducts ongoing research and actively delivers content that may interest users, evolving from reactive tool to proactive research partner. We look forward to DeepResearch creating substantial value in high-impact application scenarios, from academic research to business intelligence and policy analysis.

## 5 CONCLUSION

This paper introduces DEEPRESEARCH-REPORTEVAL, a comprehensive evaluation framework that addresses the critical challenge of assessing DeepResearch systems through their most representative outputs: research reports. Our framework systematically evaluates reports across three essential dimensions: quality, redundancy, and factuality. Through iterative LLM-human alignment, we achieve strong concordance with expert judgments, demonstrated by low MAD values and 61.11% ranking agreement. Evaluation of four leading commercial systems reveals distinct design philosophies and performance trade-offs, providing valuable insights into current research automation capabilities. We contribute a curated benchmark of 100 representative queries spanning 12 categories reflecting real-world usage patterns. As DeepResearch evolves from information assistants toward research partners, our framework provides tools for measuring progress and guiding development in this field.

## 6 ETHICS STATEMENT

This work involves human experts scoring reports. No sensitive data was shared with annotators, and no sensitive data will be included in the public release. All open-source queries are fully rewritten and reviewed by humans. All prompts used in this study are publicly released. The study poses no privacy or ethical risks.

## 7 REPRODUCIBILITY STATEMENT

We will open-source all the prompts, data, and code we use.

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

# Appendix

## A  EXAMPLES FROM DEEPRESEARCH-REPORTEVAL

Table A1 presents additional query examples from DEEPRESEARCH-REPORTEVAL, covering diverse categories. These are classical DeepResearch queries that are complex and open-end.

Table A1: Representative queries from DEEPRESEARCH-REPORTEVAL of 12 categories.

| Category | Query |
|---|---|
| Science & Technology | The current research and industrialization status of solid-state lithium batteries, including their theoretical and practical advantages compared to traditional batteries, the main challenges faced in laboratory research and mass production, the estimated timeline for application, and other potential competing technologies. |
| Health & Medicine | What are the main differences between sertraline and escitalopram? Why would a doctor choose to prescribe one medication over the other based on the patient's condition? Please compare the two in terms of their effects on causing drowsiness and nausea. |
| Economy & Business | Develop a framework or summary of economic turning points and their driving factors, including leading and lagging indicators. |
| Politics & Society | Analyze the relationship between gaze and power based on Foucault's theory of panoramic prison, elucidate their interactions, and delve deeply into the illusion of freedom. |
| History & Culture | What are the main schools of thought in the study of intellectual history? What are the characteristics of their methods in studying intellectual history? |
| Art, Music & Literature | Provide a detailed study of the personal life and work of French composer and pianist Cécile Chaminade, including her anecdotes and true stories. |
| Entertainment & Fashion | Investigate the history of the 'Zelda' series and explain the story of 'Tears of the Kingdom' in a simple and easy-to-understand way for those who are unfamiliar with gaming history, making it comprehensible even to non-fans. |
| Sports & Fitness | Since the construction of the wind tunnel, McLaren F1 team's aerodynamic design level has continuously improved. Why has such a significant breakthrough been achieved in just a few years? By the F1 races up to 2025, McLaren cars have demonstrated a clear advantage compared to other teams, even surpassing the competitiveness of Red Bull Racing's Max Verstappen in his Red Bull car. Analyze the reasons for McLaren's success and its advantages in specific aspects. |
| Education | Analyze the practice-oriented activities of communicative general learning activities for students in the basic education stage based on the collective academic dialogue techniques of literature reading classes. |
| Environment & Nature | What is the secret of the Bermuda Triangle and the reason for its mystery? Analyze the unique current systems, magnetic anomalies, and meteorological conditions in this sea area. |
| Lifestyle | Design a 7-day itinerary for Banff and the surrounding areas in Canada. On the first day, depart from Toronto to Calgary and stay one night in Calgary; the rest of the itinerary can be arranged freely. |
| Other | Investigate whether Yordles in League of Legends truly die when they suffer fatal damage, i.e., whether Yordles possess immortality characteristics, and if immortality does exist, what its specific mechanism of operation is. |

# B    PROMPTS FOR LLM-AS-A-JUDGE EVALUATION

## B.1    QUALITY

---

**Quality Judge**

You are a professional text quality evaluation expert. I will provide you with a research question and a report, please evaluate the quality of the report.

Please objectively evaluate the quality of the following report based on these four criteria, and provide scores out of 4 for each:

(1) Comprehensiveness and Depth: The report should cover all key aspects of the topic, with detailed and in-depth analysis.

(2) Structural Clarity and Logic: The report should be well-organized, with clear structure and logical flow.

(3) Fluency and Consistency of Expression: The language should be fluent, accurate, and professional, with a consistent style.

(4) Material Integration and Originality: The report should demonstrate independent thinking, effective integration of materials, and avoid simple patchwork or direct copying.

Overall Report Scoring (0–4 Points)

1. Comprehensive and In-depth Content

| Score | Description |
|——-|————-|
| 4 | The content is comprehensive, all necessary elements are present, information is detailed, analysis is in-depth, viewpoints are profound, and there are no significant omissions. |
| 3 | The content is relatively complete, most key elements are included, the analysis has some depth, and minor details are lacking but do not affect overall understanding. |
| 2 | Some content is missing, important information is not covered, the analysis is superficial, and this affects a comprehensive understanding of the topic. |
| 1 | The content is seriously incomplete, key elements are missing, the analysis is shallow, and the information is insufficient to support the topic. |
| 0 | The content is extremely lacking, there is almost no valid information, no depth at all, and it cannot form an effective report. |

2. Clear and Logical Structure

| Score | Description |
|——-|————-|
| 4 | The structure is clear and logical, each part is well-organized, transitions are smooth, and the report is easy to understand. |
| 3 | The structure is generally reasonable and logical, but some paragraphs are slightly disorganized or transitions are awkward. |
| 2 | The structure is rather loose, lacking natural segmentation and clear transitions, and the logical relationships are not clear enough. |
| 1 | The structure is chaotic, with large logical leaps, making it difficult for readers to follow the train of thought. |
| 0 | There is no discernible structure, the content is arranged chaotically, and the logic is completely unclear. |

---

3. Fluent and Consistent Expression

| Score | Description |
|——-|————-|
| 4 | The language is fluent and accurate, the style is consistent and professional, with no redundancy or improper word usage. |
| 3 | The expression is generally clear, the writing style is mostly consistent, with occasional repetition or lack of conciseness. |
| 2 | There is considerable redundancy or improper word usage, sentences are not smooth, and the style is somewhat inconsistent. |
| 1 | The language is confusing, with serious repetition, and the writing style is chaotic. |
| 0 | The expression is extremely poor, comprehension is affected, there are frequent grammatical errors, and it is difficult to read. |

4. Material Integration and Originality

| Score | Description |
|——-|————-|
| 4 | The content is highly original, with almost no material patchwork; analysis is based on a deep understanding and reprocessing of materials, and independent insights are prominent. |
| 3 | Independent analysis is predominant, material patchwork is rare, and cited materials are effectively integrated and processed. |
| 2 | Material patchwork is obvious, analysis mostly relies on direct quotations, and there is a lack of in-depth processing and independent insights. |
| 1 | The content mainly consists of simple listing of materials, with little analysis; most content is directly excerpted, and personal thinking is almost absent. |
| 0 | The entire report is a patchwork or direct copy of materials, with no analysis at all; the content is mechanically pieced together and completely lacks originality. |

5. Overall, do you like this report? Why?

| Score | Description |
|——|————-|
| 4 | Like it very much. |
| 3 | Like it. |
| 2 | It's average. |
| 1 | Don't like it much. |
| 0 | Don't like it at all. |

Notes: - A satisfactory performance deserves around 2 points, with higher scores for excellence and lower scores for deficiencies. - You should not easily assign scores higher than 3 or lower than 1 unless you provide substantial reasoning.

- Please provide the final scores in the following JSON format:

{{

"Reason": <reasoning for the scores>,

"Comprehensiveness_Score": <score>,

"Coherence_Score": <score>,

"Clarity_Score": <score>,

"Insightfulness_Score": <score>,

"Overall_Score": <score>

}}

Research Question: {question}

Research Paragraphs: {paragraph}

## B.2 REDUNDANCY

**Redundancy Judge**

Given two paragraphs, please assess the degree of content repetition between them. You should analyze from multiple perspectives and assign a reasonable score based on the scoring criteria.

# What is "Repetition":

Repetition of viewpoints or content: The two paragraphs express the same or highly similar viewpoints, themes, or conclusions, regardless of whether they are rephrased.

Repetition of examples, data, or references: The same cases, data, facts, or sources are used, or the same content is rephrased or paraphrased.

Implicit repetition: Although the wording is different, the core information, arguments, or conclusions are essentially the same.

# What is NOT "Repetition":

Differences in expression: Only the language, sentence structure, or style is different, but the information content and core viewpoints are not the same.

Related topics but different content: The topics are similar, but the information, arguments, or conclusions conveyed are different.

Supplementation and extension: One paragraph supplements, expands upon, or introduces new viewpoints to the other, rather than simply repeating it.

# Notes

Focus on content: Concentrate on repetition at the information level, not just superficial language or stylistic differences.

Consider both explicit and implicit repetition:

Explicit repetition: Direct copying or nearly identical text.

Implicit repetition: Expressing the same information through paraphrasing, summarizing, etc.

Consider contextual impact: Assess whether the repetition affects readability and information density.

Avoid subjective bias: Do not rely on personal knowledge or judgments about the correctness of the content; score only based on whether repetition exists between the paragraphs.

# Scoring Criteria

Use a 0–4 point scale to evaluate the degree of repetition between paragraphs:

4 points (Almost no repetition): The paragraphs are completely independent, with no repeated viewpoints, examples, or expressions.

3 points (Slight repetition): There are 1–2 minor instances of content repetition, but they do not affect the overall reading experience.

2 points (Some repetition): There are multiple instances of content repetition, which somewhat affect the reading experience.

1 point (Severe repetition): There is a large amount of content repetition, which seriously affects the quality of the writing.

0 points (Excessive repetition): Almost all content is repeated, and the value of the writing is lost.

# Important Notes:

- Please do not rely on your external knowledge; make judgments solely based on the provided content

- Note that using the same example to explain different concepts is not considered repetition

# Some tips may help you:

- Check if the paragraphs contain the same quotations

- Check if the paragraphs follow the same logical flow

# Some Examples

Paragraph 1:

Teamwork is essential for achieving organizational goals. When team members collaborate, they can share ideas, solve problems together, and increase productivity. Effective teamwork also improves communication and builds trust among members, which leads to a more harmonious work environment. Without teamwork, organizations may struggle to reach their objectives efficiently.

Paragraph 2:

Teamwork is essential for achieving organizational goals. When team members collaborate, they can share ideas, solve problems together, and increase productivity. Effective teamwork also improves communication and builds trust among members, which leads to a more harmonious work environment. Without teamwork, organizations may struggle to reach their objectives efficiently.

Output:

{{

"score": 0,

"explanation": "The two paragraphs are completely identical in content, wording, and structure. Every viewpoint, example, and conclusion is repeated without any difference.",

"repetitions_found": [

"Teamwork is essential for achieving organizational goals.",

"When team members collaborate, they can share ideas, solve problems together, and increase productivity.",

"Effective teamwork also improves communication and builds trust among members, which leads to a more harmonious work environment.",

"Without teamwork, organizations may struggle to reach their objectives efficiently."

],

"confidence": "100%"

}}

Paragraph 1:

Teamwork is essential for achieving organizational goals. When team members work together, they can share ideas, solve problems collectively, and increase overall productivity. Furthermore, effective teamwork enhances communication and builds trust among colleagues, creating a positive and supportive work environment. Organizations that lack teamwork often face challenges in reaching their targets efficiently.

Paragraph 2:

Achieving organizational goals greatly depends on teamwork. By collaborating, team members are able to exchange ideas, address challenges as a group, and improve productivity. In addition, good teamwork fosters better communication and trust, which are crucial for a harmonious workplace. Without strong teamwork, organizations may find it difficult to accomplish their objectives.

Output:

{{

"score": 1,

"explanation": "Most of the content is repeated between the two paragraphs. The main viewpoints, examples, and conclusions are the same, with only minor differences in wording.",

"repetitions_found": [

"Teamwork is essential for achieving organizational goals / Achieving organizational goals greatly depends on teamwork.",

"Team members share ideas, solve problems together, and increase productivity.",

"Effective teamwork improves communication and builds trust among colleagues, creating a positive work environment.",

"Organizations without teamwork struggle to reach their objectives."

],

"confidence": "95%"

}}

Paragraph 1:

Teamwork plays a vital role in helping organizations reach their goals. When individuals collaborate, they can share their unique perspectives and solve problems more effectively. Teamwork also helps distribute the workload evenly, preventing burnout and ensuring that tasks are completed on time. Moreover, working in teams can inspire creativity and innovation, as members build on each other's ideas.

Paragraph 2:

The importance of teamwork in organizations cannot be overstated. By working together, employees can share ideas and solve problems as a group, which often leads to better solutions. Additionally, teamwork can improve job satisfaction and foster a sense of belonging among employees. When people feel they are part of a team, they are more likely to be motivated and committed to their work.

Output:

{{

"score": 2,

"explanation": "There are multiple instances of content repetition, such as sharing ideas and solving problems together, but each paragraph also contains unique points (e.g., preventing burnout, inspiring creativity, job satisfaction, sense of belonging).",

"repetitions_found": [

"Teamwork helps organizations/employees share ideas and solve problems together."

], "confidence": "90%"

}}

Paragraph 1:

Participating in a marathon is an excellent way to improve one's physical endurance. For instance, both the Boston Marathon and the New York City Marathon require runners to train for months, gradually increasing their running distance and stamina. Compared to local 5K races, these world-famous marathons demand a much higher level of physical preparation, pushing athletes to reach new heights in cardiovascular fitness and muscle strength. This highlights how marathon events serve as a benchmark for testing and enhancing physical capabilities.

Paragraph 2:

Marathon running is also a powerful tool for building mental resilience. Take the Boston Marathon as an example: while the event is renowned for its challenging course, what truly sets it apart is the psychological battle runners face, especially when tackling the infamous Heartbreak Hill. Unlike local 5K races, where the mental challenge is relatively minor, marathon participants must overcome self-doubt and fatigue over several hours. This demonstrates that marathons not only test the body but also cultivate perseverance and mental toughness.

Output:

{{

"score": 3,

"explanation": "There is only slight repetition. Both paragraphs use the example of the Boston Marathon, but Paragraph 1 focuses on physical endurance and training, while Paragraph 2 emphasizes mental resilience and psychological challenges.",

"repetitions_found": [ "Both paragraphs mention the Boston Marathon as a key example." ],

"confidence": "85%"

}}

Paragraph 1:

Teamwork fosters creativity by bringing together people with diverse backgrounds and skill sets. When individuals with different perspectives collaborate, they can generate innovative solutions that might not have been possible working alone. This diversity of thought is a key driver of progress and adaptability in today's fast-changing business environment.

Paragraph 2:

Teamwork also plays a crucial role in conflict resolution within organizations. When disagreements arise, a strong team can address issues openly and constructively, ensuring that all voices are heard and a consensus is reached. This ability to manage and resolve conflicts effectively contributes to a healthier and more productive workplace.

Output:

{{

"score": 4,

"explanation": "The two paragraphs are completely independent. One discusses creativity from diverse backgrounds, the other focuses on conflict resolution. There are no repeated viewpoints, examples, or expressions.",

"repetitions_found": [],

"confidence": "100%"

}}

# Output Format

Must output a JSON object with the following fields:

{{

"score": "0-4 score based on above criteria",

"explanation": "Explanation of score with specific examples of repetition",

"repetitions_found": [the repeated content1,the repeated content2,the repeated content3,...],

"confidence": "Confidence in assessment (0%-100%)"

}}

# Output Example:

{{

"score": 3,

"explanation": "Score of 3 given due to low overall repetition, with only one unnecessary repetition in describing [concept].",

"repetitions_found": [the repeated content1,the repeated content2,the repeated content3,...],

"confidence": 90

}}

Please evaluate the degree of repetition between paragraphs in the following <paragraphs>.

Paragraph 1:

{para1}

–paragraph1 end–

Paragraph 2:

{para2}

–paragraph2 end–

output:

## B.3 FACTUALITY

**Factuality Judge**

# Factual Evaluation
Given a piece of web content and a sentence from a report, please determine whether the information expressed in the sentence can be directly found in or reasonably inferred from the provided web content.
# Important Notes
- Please do not rely on your external knowledge; make judgments solely based on the provided web content
- Pay attention to key terms in the statement (such as time, location, people, quantities, etc.), ensuring these details have corresponding or derivable information in the web content
- If the statement contains multiple information points, please evaluate each one to determine if they can all be supported by the web content
## Scoring Criteria

- **Fully Supported**:
- The web content explicitly mentions information that is identical to or highly relevant to the statement, allowing direct verification of the statement as true
- Or, through reasonable inference from multiple information points in the web content, a conclusion consistent with the statement can be reached
- **Partially Supported**:
- The web content contains some relevant information, but it is insufficient to fully confirm or deny the statement
- Or the information is ambiguous, making it impossible to make a clear judgment
- **Not Supported**:
- The web content does not mention any information related to the statement
- Or the web content clearly contradicts the statement, allowing the statement to be determined as false
Please return the analysis result in JSON format:
{{
"is_factual": -1/0/1, # -1: not supported, 0: partially supported, 1: fully supported
"sentence_support": "Specific sentences from the web content that can support this fact"
}}
Here is the content of the website: {markdown}
Here is the sentence: {input}

## C  RELATED WORK

**DeepResearch Systems.**    DeepResearch systems have gained significant attention for their ability to conduct expert-level research in knowledge-intensive scenarios. On the open-source front, advanced projects (Tang et al., 2025; Hu et al., 2025; Fang et al., 2025; MiroMind AI Team, 2025; LangChain & contributors., 2025) demonstrate steady progress in research automation and collaborative knowledge construction. In parallel, leading commercial products (Google, 2025; OpenAI, 2025; Qwen, 2025; Perplexity, 2025) exhibit strong performance in complex real-world tasks while maintaining rapid iteration cycles. Collectively, these developments are transforming DeepResearch from an "information assistant" into a true "research partner" for high-value applications.

**Knowledge-Enhanced LLM Agents.**    To enable agents to acquire external knowledge, early approaches (Qin et al., 2023) employed supervised fine-tuning on training data containing search tool calls. With the emergence of retrieval-augmented generation (RAG) techniques (Guo et al., 2024; Qian et al., 2024; Allahverdiyev et al., 2024; Gutiérrez et al., 2024; Fan et al., 2025), well-engineered RAG pipelines became a prevalent approach for models to access external knowledge through prompt construction and LLM invocation. More recently, reinforcement learning (RL) methods have demonstrated superior generalization capabilities, leading agent training to adopt RL approaches for improving real-world performance (Jin et al., 2025; Shi et al., 2025; Wu et al., 2025a; Zheng et al., 2025; MiroMind AI Team, 2025; Tongyi DeepResearch Team, 2025). These agents exhibit advanced capabilities including multi-step web browsing, information synthesis, and iterative query refinement.

**Evaluation of DeepResearch.**    The complexity of DeepResearch systems makes their evaluation particularly challenging. Benchmarks based on RAG (Yang et al., 2018; Joshi et al., 2017; Kočiský et al., 2017) or complex web search (Wei et al., 2025; Chen et al., 2025) are insufficient for assessing the comprehensive outputs of DeepResearch systems. Since DeepResearch reports emphasize both content quality and evidence traceability, general writing benchmarks (Bai et al., 2024; Wu et al., 2025b) are likewise inadequate. Recent work has introduced specialized benchmarks targeting DeepResearch capabilities. For instance, DeepResearch Bench (Du et al., 2025) compares system outputs against predefined reference reports, while ResearcherBench (Xu et al., 2025) employs expert-designed criteria to evaluate insight quality. Nevertheless, there remains a critical need for benchmarks that are more generalizable, practical, and straightforward to implement.

# D    THE USE OF LARGE LANGUAGE MODELS (LLMs)

We use LLMs as evaluation tools in our experiments. We employed LLMs to assist in revising a very small number of sentences in the paper.

