# OpenReview forum: "Understanding DeepResearch via Reports"
_ICLR.cc/2026/Conference — Submitted to ICLR 2026_

### Official Review · Reviewer_drL7 · 2025-10-27

**Soundness:** 2
**Presentation:** 2
**Contribution:** 1
**Rating:** 2
**Confidence:** 3

**Summary:**

This paper introduces a benchmark and an evaluation pipeline for assessing DeepResearch systems in the research report generation task. The proposed benchmark consists of 100 queries across 12 categories. The evaluation pipeline is based on LLM-as-a-judge, with prompts tuned through a manual adjustment framework inspired by TextGrad. The authors evaluate DeepResearch in terms of the quality, redundancy, and factuality of the generated reports and claim that their LLM-as-a-judge-based pipeline achieves reasonable alignment with human evaluation, with 61.11% agreement in the task of ranking three reports.

**Strengths:**

* This paper introduces a new dataset consisting of 100 queries across 12 categories for evaluating DeepResearch on the report generation task.

* They propose a framework to manually improve prompts for LLM-as-a-judge, which is inspired by TextGrad. The idea is not academically novel, but the framework can be practically useful.

**Weaknesses:**

The proposed dataset is not substantially novel compared to existing datasets such as [1]. Therefore, I consider the evaluation pipeline to be the main contribution. However, I am not fully convinced that the proposed evaluation framework is substantially novel or offers clear advantages over existing methods.

* The rationale behind the criteria selection and categorization is unclear. In particular, Quality includes Comprehensiveness, Coherence, Clarity, and Insightfulness, which are largely different. It is unclear why Redundancy and Factuality are specifically isolated from these criteria.

* Doubt on the reliability of LLM-as-a-judge. I agree that redundancy and factuality can be evaluated without references. However, without prior information such as reference reports, it is not persuasive to assume that LLM-as-a-judge can reliably evaluate comprehensiveness, which is one of the important criteria. For such criteria, reference-based evaluation [1] would be more reasonable and reliable. This concern is consistent with the result in Table 1, which shows that the Quality MAD is relatively high.

* Novelty of the evaluation pipeline is not substantial. The proposed pipeline represents a standard and straightforward application of LLM-as-a-judge.

[1] Du et al. (2025). DeepResearch Bench: A Comprehensive Benchmark for Deep Research Agents. https://arxiv.org/abs/2506.11763.

**Questions:**

I expect responses to the points listed in the Weaknesses section.

---

> ### Author Response · Authors · 2025-11-20
>
> Dear Reviewer drL7,
>
> We thank you for your positive feedback that our work "practically useful". Below are our responses to your questions, and we hope they help alleviate your concerns.
>
> ---
>
> Q1.The rationale behind the criteria selection and categorization is unclear. In particular, Quality includes Comprehensiveness, Coherence, Clarity, and Insightfulness, which are largely different. It is unclear why Redundancy and Factuality are specifically isolated from these criteria.
>
> A1. Because quality is complex and subjective, it is difficult to standardize its evaluation, whereas redundancy and factuality can be assessed through **carefully designed tasks that reduce reliance on LLM-as-a-judge.**
> Quality, as a **high-level, integrative dimension**, encompasses sub-aspects such as comprehensiveness, coherence, clarity, and insightfulness. These components heavily depend on evaluators’ subjective understanding, making them difficult to fully disentangle or replace through task design or automated metrics. In other words, the quality dimension reflects a comprehensive perception of a report’s overall value, and its assessment cannot be easily standardized via simplified tasks.
> In contrast, **redundancy and factuality are more operationalizable and locally observable**. For example, redundancy can be quantified through pairwise segment comparisons, and factuality can be verified against external knowledge sources. Such properties allow our task design to mitigate direct dependence on LLM-as-a-judge for these two dimensions.
>
>
> ---
>
> Q2. Doubt on the reliability of LLM-as-a-judge. I agree that redundancy and factuality can be evaluated without references. However, without prior information such as reference reports, it is not persuasive to assume that LLM-as-a-judge can reliably evaluate comprehensiveness, which is one of the important criteria. For such criteria, reference-based evaluation [1] would be more reasonable and reliable. This concern is consistent with the result in Table 1, which shows that the Quality MAD is relatively high.
>
> A2. Ideally, having reference reports that serve as golden answers would indeed be preferable. However, **this is practically infeasible**. Research reports rarely have a single “golden answer”, and constructing high-quality human-authored references is prohibitively expensive. Moreover, using LLM-generated reports (like [1]) as references would risk compromising evaluation quality due to potential flaws in the generated content.
> Additionally, LLM-as-a-judge has become a common approach in long-text evaluation (e.g., EQ-Bench  [2] used in Grok-4.1 evaluations). With carefully designed prompts and calibration mechanisms, LLM-as-a-judge can provide reasonable assessments even in the absence of reference texts. Given the infeasibility of obtaining reliable references, this strategy represents a pragmatic trade-off between scalability and evaluation validity.
>
> [1] DeepResearch Bench: A Comprehensive Benchmark for Deep Research Agents. https://arxiv.org/abs/2506.11763.
>
> [2] EQ-Bench: An Emotional Intelligence Benchmark for Large Language Models. https://arxiv.org/abs/2312.06281.
>
> ---
>
> Q3.Novelty of the evaluation pipeline is not substantial. The proposed pipeline represents a standard and straightforward application of LLM-as-a-judge.
>
> A3. Our core contribution lies in demonstrating that, **through thoughtful task design (e.g., multi-dimensional and orthogonal metrics) and calibration mechanisms (e.g., triplet ranking), making a standard LLM-as-a-judge approach effectively align with human evaluations in the DeepResearch setting**. Given the current lack of established evaluation protocols for DeepResearch Domain, this reproducible, scalable, and human-aligned evaluation pipeline itself holds practical value.
>
>
>
> ---
>
> If you have any further concerns, we are happy to discussing them with you. We hope our responses have addressed your questions. If so, would you consider raising your score accordingly?

---

> > ### Comment · Reviewer_drL7 · 2025-11-20
> >
> > Thank you for your response. Since it does not address the major concerns outlined in my original review, I will maintain my initial score.
> >
> > ### Q1
> >
> > I agree that the quality is "difficult to fully disentangle or replace through task design or automated metrics". However, your evaluation framework does introduce automated metrics for quality as described in Section 2.3.1. The paper should clearly explain and justify why this proposed design is reasonable.
> >
> > ### Q2
> >
> > > With carefully designed prompts and calibration mechanisms, LLM-as-a-judge can provide reasonable assessments even in the absence of reference texts
> >
> > If you make this claim, the work should include a comparison between evaluations conducted with and without reference texts. From Table 1, I am left with the impression that the proposed evaluation framework is not necessarily reliable.
> >
> > ### Q3
> >
> > > making a standard LLM-as-a-judge approach effectively align with human evaluations in the DeepResearch setting
> >
> > I respectfully disagree that this work is properly designed to support this claim. As noted in my responses to Q1 and Q2, the paper does not provide a clear explanation of the evaluation framework, nor does it present experimental analyses demonstrating that the proposed framework is superior to alternative approaches.

---

### Official Review · Reviewer_2rKL · 2025-10-29

**Soundness:** 2
**Presentation:** 3
**Contribution:** 1
**Rating:** 2
**Confidence:** 4

**Summary:**

This paper introduces a framework for evaluating  deep research systems by producing long research-style reports. The framework measures quality, redundancy, and factuality using an LLM-as-a-Judge. They benchmarked four commercial systems across 100 queries from 12 categories. They report low MAD between LLM and human scores  to show that their evaluation aligns with human evals.

**Strengths:**

- very timely problem
- studying tradeoffs between characteristics of reports vs evaluation metrics
- Tend to also include  llm-human eval agreement
- promised for public release of prompts and benchmark queries
- easy to read and follow

**Weaknesses:**

- The restriction to three metrics (quality, redundancy, factuality) omits other critical aspects such as relevance, novelty, interpretability, etc. I think it would be more exciting to focus on more novel aspects of evaluation.

- I might have missed it but it was not clear what is the soruce of 150 k real-world queries, how did they classify them?  some more details about the benchmark would be appreciated.

- I could not understand exactly how the 61.11 % agreement between llm and human was measured. If it is pairwise agreement, then random baseline would give 50% and thus 61% is not really significant agreement

- I am not sure that using MAD between LLM and human scores is the best way to calibrate alignment between human and model ratings. This metric emphasizes absolute score matching rather than relative ranking consistency, which might be the more meaningful signal in evaluation. For instance, the LLM and human annotators may operate on different scoring scales; yet, if their relative ordering of reports aligns well, that would better reflect evaluation reliability. In contrast, a system with a constant absolute deviation (e.g., always two points higher or lower than humans) could still achieve low MAD but fail to capture true ranking agreement.

- I think pairwise redundancy scoring can be exploited easily e.g., a nonsensical report with unrelated paragraphs could appear perfectly non-redundant.

- The factuality evaluation only tests claim–source overlap; it does not check whether unsupported claims exist, nor citation precision.
Please check the following papers which have more accurate explanation of citation precision and coverage.
https://arxiv.org/abs/2411.17375
https://arxiv.org/abs/2304.09848

- some open-source and task-specific DeepResearch frameworks are missing for example
-- Deepresearcher
- openscholar https://arxiv.org/abs/2411.14199
- LLMs + web search for example https://github.com/sentient-agi/OpenDeepSearch

- The framework’s calibration is dataset-specific; there is no evidence it generalizes to unseen queries. It would be useful to show how would calibration before and after alignment changes between llm and human annotation.

-  Section 4 provides general reflections  which could be briefly mentioned in intro. I did not find section 4 informative or necessary to keep in the main part of the paper

**Questions:**

- I am not sure if I missed this but how were the 150 k real-world queries obtained, filtered, and categorized?
- Why were only three dimensions (quality, redundancy, factuality) chosen? Why not additional ones  like relevance, novelty, etc?
- Who were the three human annotators?were they domain experts or crowdworkers? Since some topics are very niche, If they are not expert, it might not be a good idea to calibrate llm as a judge with their opinion.
- In page 5, what agreement metric does the 61.11 % refer to? Pairwise accuracy, Kendall tau? simple agreement? if it is simple agreement between 2 options, then the random results will lead to 50% agreement and then 61% is not really impressive.
- How does the calibration process generalize beyond the 120 aligned reports? how did the scores changed before and after calibration?
- Is redundancy metric gamifiable? If I have  non-coherent but dissimilar text (high non-redundancy but low quality)?

---

> ### Author Response · Authors · 2025-11-20
>
> Dear Reviewer 2rKL,
>
> We thank you for your positive feedback that our work "easy to read and follow" and consider it as a "very timely problem". Below are our responses to your questions, and we hope they help alleviate your concerns.
>
> ---
>
> Q1. How were the 150k real-world queries obtained, filtered, and categorized?
>
>
> A1. These 150k real-world queries originate from a publicly deployed DeepResearch system and reflect authentic user interaction behaviors. Due to anonymity considerations, we are currently unable to disclose specific details about the data source. We commit to publicly releasing the source information if this paper is accepted.
>
> ---
>
>
> Q2. Why were only three dimensions (quality, redundancy, factuality) chosen? Why not additional ones like relevance, novelty, etc?
>
> A2. We selects these three core dimensions (quality, redundancy, and factuality) to capture the essential value of research reports: content depth (quality), expressive efficiency (redundancy), and objective accuracy (factuality). These aspects represent the most critical and challenging elements in evaluating research reports and also offer relatively clear criteria for measurement, facilitating quantifiable or semi-automated assessment in the absence of external references (e.g., human ratings or other references).
>
> ---
>
> Q3.Who were the three human annotators? were they domain experts or crowdworkers? Since some topics are very niche, If they are not expert, it might not be a good idea to calibrate llm as a judge with their opinion.
>
> A3. The three human annotators are all university graduates with strong general comprehension and critical reasoning skills. Given the breadth of research topics covered, it was impractical to recruit domain experts for every relevant field. Instead, we selected annotators with solid educational backgrounds and strong analytical abilities, and ensured the reliability of their judgments through detailed annotation guidelines and a rigorous calibration process. Their general cognitive competence is sufficient to carry out the evaluation tasks effectively.
>
> ---
>
>
> Q4.In page 5, what agreement metric does the 61.11 % refer to? Pairwise accuracy, Kendall tau? simple agreement? if it is simple agreement between 2 options, then the random results will lead to 50% agreement and then 61% is not really impressive.
>
> A4. As stated in lines 350–354 of the paper, the reported 61.11% refers to the exact match accuracy for full triplet rankings, not the accuracy under a binary choice setting. Specifically, for 120 queries, each query yielded three reports generated by different DeepResearch systems, resulting in a total of 360 reports. For each query, human experts provided a complete total ordering of the three reports (i.e., explicitly ranking them as 1st, 2nd, and 3rd). Our system also assigned scores to these reports and produced its own ranking based on those scores. In evaluation, a prediction was counted as correct only when the system’s full ranking matched the human expert’s ranking. Under this setting, the expected accuracy of a random baseline is 1/6, not 50%. Therefore, achieving an exact match accuracy of 61.11% demonstrates significant performance on this challenging ranking task.
>
> ---
>
> Q5.How does the calibration process generalize beyond the 120 aligned reports? how did the scores changed before and after calibration?
>
>
> A5. Our original wording may have caused some confusion. The calibration phase and the validation phase use different sets of 120 reports. In Sec 2.3.4, we evaluate the calibration effectiveness of the proposed judge method in two distinct phases (as shown in Fig. 4):
> 1. Calibration Phase: We calibrate the judge method using 120 reports (corresponding to 120 distinct queries), with the goal of aligning its scores as closely as possible with human judgments. After calibration, as shown in Table 1, the method achieves a low MAD on these reports.
> 2. Validation Phase: To assess the generalization capability of the calibrated judge, we collect a completely new set of reports. For each query, we generate one report using each of three different DeepResearch systems, resulting in 120 × 3 = 360 reports in total. We then:
> -  Apply the calibrated judge to automatically score and rank the three reports within each query group;
> -  Independently obtain human expert rankings for the same triplets of reports.
> The agreement between the automatic rankings and human rankings serves as the validation metric for calibration effectiveness, yielding an exact-match accuracy of 61.11%. This result demonstrates that our judge method, once calibrated, exhibits strong rank-preserving capability on unseen data.

---

> ### Author Response · Authors · 2025-11-20
>
> Q6.Is redundancy metric gamifiable? If I have non-coherent but dissimilar text (high non-redundancy but low quality)?
>
>
> A6. Yes, **the redundancy  itself focuses solely on textual repetition or semantic overlap** and does not directly reflect coherence or overall quality. Therefore, **we include two additional, orthogonal dimensions—quality and factuality**.
> As you rightly point out, **it is possible for a system to achieve “high non-redundancy but low quality”**: for instance, generating text that lacks coherence, yet appears non-redundant because its segments differ substantially from one another. This is precisely why our evaluation framework explicitly incorporates three orthogonal dimensions—quality, redundancy, and factuality. We fully agree that not all DeepResearch systems can simultaneously excel across all dimensions. For example, in our experiments (see lines 384–386), we observed that **perplexity often produce reports with extremely low redundancy, yet these reports suffer from poor logical coherence and significantly lower overall quality—exactly matching the scenario you described.**
> Consequently, we do not treat quality or any other single dimension as the sole or absolute criterion for evaluation. Given that no ideal DeepResearch system currently exists, diverse output paradigms hold practical value: on one hand, low-redundancy, high-information-density reports with limited quality may suit scenarios prioritizing broad coverage; on the other hand, slightly redundant but high-quality outputs may be preferable for tasks demanding high reliability. This multi-dimensional trade-off enables a more comprehensive characterization and guidance of DeepResearch system development.
>
>
> ---
>
> If you have any further concerns, we are happy to discussing them with you. We hope our responses have addressed your questions. If so, would you consider raising your score accordingly?
>
> Best,
>
> Authors

---

### Official Review · Reviewer_oJ16 · 2025-10-31

**Soundness:** 3
**Presentation:** 3
**Contribution:** 3
**Rating:** 6
**Confidence:** 2

**Summary:**

The paper presents a framework for evaluating AI research agents based on their final research reports instead of single tasks. It measures three aspects, quality, redundancy, and factuality, using LLM-as-a-Judge approach aligned with human experts. The authors benchmark four commercial systems (OpenAI, Perplexity, Gemini, Qwen) on 100 real-world research queries across 12 categories. Results show that Qwen and OpenAI perform best overall, producing reports that are more coherent, factually grounded, and insightful.

**Strengths:**

- The paper addresses the growing need to evaluate full research-capable AI systems, not just text generators or search agents.
- The three dimensions (quality, redundancy, factuality) give a clear and balanced way to judge complex research reports.
- The authors test multiple commercial DeepResearch systems on real-world queries covering 12 diverse categories, providing rich insights.
- The paper discusses trade-offs (e.g., length vs. clarity) and highlights distinct design philosophies among systems, showing awareness of practical challenges.

**Weaknesses:**

- The framework only evaluates final reports and does not analyze intermediate reasoning steps or tool usage during the research process.
- The evaluation focuses on AI-related and factual domains; it’s unclear how well it applies to creative or interdisciplinary research.
- The framework doesn’t capture how agents perform over multiple iterations or collaborations with human researchers.

**Questions:**

- Real research often involves iteration and feedback. Could the framework be extended to evaluate multi-turn or collaborative research settings, where AI systems revise their reports based on critique or new data?
- The framework focuses mainly on assessing the final reports. It would be interesting to also evaluate intermediate steps, such as how the systems plan, search, and integrate evidence during the research process.

---

> ### Author Response · Authors · 2025-11-20
>
> Dear Reviewer oJ16,
>
> We sincerely appreciate for your positive feedback that our work “give a clear and balanced way to judge complex research reports”, “providing rich insights” and “showing awareness of practical challenges”. Below are our responses to your questions, and we hope they help alleviate your concerns.
>
> ---
>
> Q1.Real research often involves iteration and feedback. Could the framework be extended to evaluate multi-turn or collaborative research settings, where AI systems revise their reports based on critique or new data?
>
> A1. Yes, absolutely. This framework can serve as an  "evaluator" for improving DeepResearch systems. Such a setup naturally supports extension to multi-turn and collaborative research scenarios. In iterative settings, one can simply feed the report generated after each iteration into ReportEval to obtain comparable scores, and further compute “iteration gains” (e.g., ΔQuality, ΔFactuality) as new metrics to comprehensively assess whether the system genuinely improves upon receiving feedback or new data. For advanced collaborative modes, such as the agent proactively querying users or experts during research, expressing uncertainty, requesting additional data, and regenerating the report, ReportEval can still serve as a unified evaluation standard for the final output.
>
> ---
>
> Q2.The framework focuses mainly on assessing the final reports. It would be interesting to also evaluate intermediate steps, such as how the systems plan, search, and integrate evidence during the research process.
>
> A2. We fully agree—this is precisely the limitation highlighted in Section 4.3 of our manuscript. Our contribution primarily lies in proposing an end-to-end evaluation framework for DeepResearch, a domain that is nearly entirely open-ended. Also, we want to call on the community to also explore process-level evaluation metrics, such as search depth, evidence diversity, planning quality, and tool-use efficiency.

---

### Official Review · Reviewer_Esec · 2025-10-31

**Soundness:** 3
**Presentation:** 3
**Contribution:** 3
**Rating:** 6
**Confidence:** 4

**Summary:**

This paper proposes DEEPRESEARCH-REPORTEVAL, a framework for evaluating DeepResearch agents through their generated research reports. It builds a benchmark of 100 real-world research queries and scores reports from four commercial systems on quality, redundancy, and factuality using an LLM-as-judge aligned with human ratings.

**Strengths:**

1, The work studied here is intereting and the conclusion is valuable for real-world applications as LLM-generated reports are expected to exist more frequently in industry.

2, It designs a somewhat reasonable scoring scheme (quality, redundancy, factuality) with LLM-human alignment, showing strong correlation with expert judgments.

3, It provides a novel realistic benchmark of 100 research queries.

**Weaknesses:**

1, Although the paper iteratively aligns LLM-as-judge with human ratings, the evaluation pipeline does not explicitly control for superficial textual cues (e.g., length, citation count, formatting quality, bullet-point structure, narrative polish) that may inflate perceived report quality.
Those info may mislead the conclusion.
2, I would recommend to measure cross-LLM specificity, i.e., whether different DeepResearch systems produce meaningfully distinct reports for the same query. This can avoid the evaluation that rewards safe, popular, and consensus-shaped outputs instead of research originality or perspective depth.

**Questions:**

What's the agreement among human evaluators, and how close the LLM-human agreement to the human-by-human ones?

---

> ### Author Response · Authors · 2025-11-20
>
> Dear Reviewer Esec,
>
> We thank you for your kind remarks that our work is “interesting”, “valuable for real-world applications”, and that we have designed a “reasonable scoring scheme”. Below are our responses to your questions, and we hope they help alleviate your concerns.
>
> ---
>
> Q1. What's the agreement among human evaluators, and how close the LLM-human agreement to the human-by-human ones?
>
>
> A1.
>
>
> The LLM-as-a-judge, after calibration with our method, achieves overall agreement with human judgments that approaches inter-human agreement. Notably, on Factuality, its consistency with humans even exceeds human-to-human consistency, demonstrating sufficient reliability.
>
>
> | Dimension     | Human–Human Consistency (MAD) | LLM–Human Consistency (MAD) |
> |---------------|-------------------------------|-----------------------------|
> | Quality       | 0.47                          | 0.72                        |
> | Redundancy    | 0.26                          | 0.31                        |
> | Factuality    | 0.36                          | 0.29 (**better than human–human**) |
>
>
> Specifically, the LLM’s agreement with humans varies by dimension: it is slightly lower than human–human agreement on Quality, comparable (though slightly worse) on Redundancy, and crucially superior to human–human agreement on Factuality. This indicates that while LLMs still exhibit a gap in evaluating highly subjective dimensions like Quality, our task decomposition and calibration approach effectively mitigates this gap for Redundancy. Moreover, on the more objective dimension of Factuality, the LLM’s judgments are even more consistent with humans than humans are with each other.

---

> > ### Comment · Reviewer_Esec · 2025-11-25
> > **Author rebuttal acknowledged**
> >
> > Author rebuttal acknowledged; my original score is fair.

---

### Meta-Review · Area_Chair_epUh · 2026-01-08

**Summary:**

The paper proposes "DeepResearch-ReportEval," a framework for evaluating DeepResearch agents based on their final output (reports). It utilizes an LLM-as-a-Judge approach calibrated to human experts to measure three dimensions: Quality, Redundancy, and Factuality. The authors benchmark four commercial systems (OpenAI, Perplexity, Gemini, Qwen) on a set of 100 queries.

**Reviewer Concerns:**

**Addressed:**

Agreement Metrics (2rKL, Esec): The authors clarified that the 61.11% agreement refers to exact matches in ranking three items (where the random baseline is 1/6, not 50%), which addresses the statistical significance concern.

Data Source (2rKL): The authors clarified that queries came from a public deployment (anonymized for review).

**Outstanding:**

validity of "Quality" Metric without References (drL7): This is the most significant outstanding issue. Reviewer drL7 rightly argues that evaluating "comprehensiveness" and "insightfulness" using an LLM without a gold-standard reference report is unreliable and prone to bias. The authors' defense—that gold standards are expensive—acknowledges the limitation but does not validate the proposed solution.

Lack of Novelty (drL7): The framework is a standard application of LLM-as-a-judge (calibrated via TextGrad-like methods). There is little algorithmic or methodological innovation.

Black-Box Evaluation (oJ16): The framework evaluates only the final PDF/text, ignoring the "Deep" aspect of the research (intermediate reasoning, tool usage, search pathing). This limits the insights regarding why a system failed.

Limited Metrics (2rKL): The restriction to only Quality, Redundancy, and Factuality omits critical aspects like relevance or novelty.

**Reviewer Scores:**

Reviewer Esec (Score: 6): Likely to stay 6. The reviewer found the work "interesting" and accepted the agreement stats, but did not strongly champion the paper against methodological critiques.

Reviewer oJ16 (Score: 6): Likely to stay 6. This reviewer noted the limitation regarding intermediate steps but felt the "end-to-end" value was sufficient for a marginal accept.

Reviewer 2rKL (Score: 2): Likely to stay 2. The reviewer raised fundamental issues with the choice of metrics and calibration (MAD vs ranking) which were not fully resolved by the rebuttal.

Reviewer drL7 (Score: 2): Likely to stay 2. The reviewer explicitly stated in the discussion that the rebuttal did not address their major concern regarding reference-free quality evaluation.

---

### Decision · Program_Chairs · 2026-01-26

Reject